

# Quantum oscillations in an impurity-band Anderson insulator

Nigel R. Cooper[1,2]⋆ and Jack Kelsall[1]

**1** Cavendish Laboratory, University of Cambridge, Cambridge, CB3 0HE, United Kingdom
**2** Department of Physics and Astronomy, University of Florence, 50019 Sesto Fiorentino, Italy

⋆ nrc25@cam.ac.uk

## Abstract

We show that for a system of localized electrons in an impurity band, which form an Anderson insulating state at zero temperature, there can appear quantum oscillations of the magnetization, i.e. the Anderson insulator can exhibit the de Haas–van Alphen effect. This is possible when the electronic band from which the localized states are formed has an extremum that traces out a nonzero area in reciprocal space. Our work extends existing theories for clean band insulators of this form to the situation where they host an impurity band. We show that the energies of these impurity levels oscillate with magnetic field, and compute the conditions under which these oscillations can dominate the de Haas–van Alphen effect. We discuss our results in connection with experimental measurements of quantum oscillations in Kondo insulators, and propose other experimental systems where the impurity band contribution can be dominant.



# 1 Introduction

One of the most striking signatures of the Fermi surface of a metal is the appearance of quantum oscillations [1]: the periodic modulation of physical observables with the inverse magnetic field. Whether seen in the magnetization (the de Haas–van Alphen effect) or in the conductivity (the Shubnikov–de Haas effect), quantum oscillations in a metal arise from the Landau quantization of the quasiparticle states close to the Fermi surface. They can be used as a highly sensitive way to measure Fermi surface properties, including the Fermi surface geometry, and the quasiparticle effective masses and scattering rates. Indeed, the requirement of small quasiparticle scattering rate for the appearance of quantum oscillations has led to their observation being taken as indicative of the existence of a Fermi liquid phase.

In recent years, this association has been called into question, in light of remarkable experimental discoveries of quantum oscillations in insulators. Quantum oscillations have been observed in the Kondo insulators $SmB_6$ [2–5] and $YbB_{12}$ [6–9], and in insulating InAs/GaSb quantum well devices [10–12]. More recently there have been reports of oscillatory behaviour in $WTe_2$ [13] and for the insulating spin-liquid $\alpha$-$RuCl_2$ [14,15], though these may be of different origin [16–18]. These discoveries have initiated a re-examination of the long-established theoretical understanding of quantum oscillations [1,19] to models that may represent these insulating materials.

One class of theories explores the effects of Landau quantization on the electronic states in band insulators. It is now understood that even very simple band insulators can give rise to de Haas–van Alphen (dHvA) oscillations [20–22]. These can arise in a class of narrow-gap insulators that have the feature that their gap minimum traces out a closed area in reciprocal space. That quantum oscillations can appear in insulators – i.e. systems without a Fermi surface – is not in conflict with standard theory for metals [1]. The amplitude of oscillations in the band insulators falls as $\exp(-B_0/B)$ as the magnetic field $B$ goes to zero, with a field scale $B_0$ that is set by the insulating gap [23]. For Fermi liquids such damping is an extrinsic effect that arises only as a result of impurities [1], so a nonzero $B_0$ in a pristine system indicates that it cannot be a metal.

Bandstructures that allow quantum oscillations in band insulators [20–22] are expected in mean-field theories of Kondo insulators [24], and are also characteristic of topological insulators [25] and two-dimensional (2D) semiconductor materials. The prevalence of these new classes of materials in modern physics makes it important to understand theories for their quantum oscillations. Recent theory has explored quantum oscillations in pristine band insulators [26–31], the roles of topological bands [21] and of impurity scattering [32,33], oscillations of the conductivity [34,35] and density of states [33], the effects of interactions [23,36–41], and the consequences of quantum spin liquids [42] and of Kondo breakdown [43]. Experimental measurements of the above Kondo insulators also show other anomalous features. The large low-temperature heat capacity has inspired theories that suggest the role of excitons [44] or impurity states [45,46].

Separate lines of theory, which can account also for the anomalous low-temperature thermal conductivity observed in Kondo insulators [4,7], postulate the existence of new phases of matter, which host Fermi surfaces for neutral fermions that can give quantum oscillations without electrical conductivity [47–53]. This remains an active and intriguing field of research, and the origin of oscillations in these materials continues to be debated.

In this paper, we extend the reach of these recent theoretical explorations by considering not a band insulator, for which the insulating behaviour is tied to the existence of an energy gap at the Fermi level, but an Anderson insulator, for which there need be no energy gap and the insulating behaviour arises from the states being spatially localized by disorder. We argue that Anderson insulators can also show quantum oscillations of the magnetization. We focus

on the magnetization, and not the conductivity, for two reasons. Firstly the magnetization is an equilibrium quantity and therefore simpler to evaluate than the response function for the conductivity. Secondly, oscillations of the magnetization can persist down to zero temperature where the system is truly insulating, whereas oscillations in the conductivity can only be considered at non-zero temperature [34, 35] where the system is no longer strictly an insulator. The model that we study starts from a narrow-gap band insulator of the form discussed previously [20], but introduces an impurity band which can host an Anderson insulating phase. Our study goes beyond previous theories of impurity scattering in narrow-gap insulators [32, 33], which treat the impurities phenomenologically through scattering rates, and which lead to a non-zero conductivity even for zero temperature. We provide a complete calculation of the energetics for a simple model of an electron bound to an impurity site. As we argue below, in the regimes in which the electrons in the impurity band form an Anderson insulator, the energetics of the impurity band system are dominated by those of the individual impurity levels. We calculate the quantum oscillations of these impurity levels, and determine the conditions under which the quantum oscillations of the impurity band can dominate those from the background band insulator. The detailed analysis is restricted to the case of dilute impurities, for which the overlap of the impurity levels can be neglected and the electrons in the impurity band are restricted to states of order the size of the bound state $a_0$. This is the limiting case of the Anderson insulator where the localization length is $a_0$. Nevertheless, it is sufficient to study this case to establish the existence of quantum oscillations in Anderson insulators. We comment below on the expected modifications in regimes in which the overlap of the impurity bound states leads to electronic states that are less strongly localized.

The paper is organized as follows. In Sec. 2 we introduce the model that we study. We then determine the properties of (shallow) bound states on impurities, in Sec. 3, and their response to external magnetic field $B$. We show that the resulting oscillations of the magnetisation are also suppressed at small magnetic fields, but now as $\exp(-B_0^{\mathrm{imp}}/B)$ with a new field scale $B_0^{\mathrm{imp}}$ for which we determine an analytic expression for our model. In Sec. 4 we discuss the experimental consequences of our results, also in connection with experimental observations of Kondo insulators. Finally in Sec. 5 we summarize our results and provide an outlook for where they are most readily observed experimentally.

## 2 Model

We consider a model of a band insulator in two dimensions formed from the hybridization of overlapping bands of spinless electrons. For simplicity of presentation the insulator that we study is non-topological. However, the features that we shall focus on – regarding the properties of impurity levels – would appear also for a topological insulator. The unhybridised bands have masses $m_1$ and $m_2$, and their energy offset leads them to cross on the circle $|\boldsymbol{p}| = p_*$, with single-particle Hamiltonian

$$\hat{H} = \begin{pmatrix} (|\hat{\boldsymbol{p}}|^2 - p_*^2)/(2m_1) & \gamma/2 \\ \gamma/2 & -(|\hat{\boldsymbol{p}}|^2 - p_*^2)/(2m_2) \end{pmatrix}. \tag{1}$$

The hybridisation $\gamma$ opens a gap close to $p_*$, which we shall assume to be small compared to the characteristic kinetic energy scales $p_*^2/(2m_{1,2})$. The resulting upper/lower bands $E^\pm(p)$ then have a band minimum/maximum at energies $E_*^\pm$ located at the momenta

$$(p_*^\pm)^2 = p_*^2 \pm \gamma \sqrt{m_1 m_2} \left( \frac{m_1 - m_2}{m_1 + m_2} \right), \tag{2}$$

which are close to $p_*$. See Fig. 1. The Fermi level is taken to lie in the gap, such that at zero temperature the lower band is filled and the upper band empty.

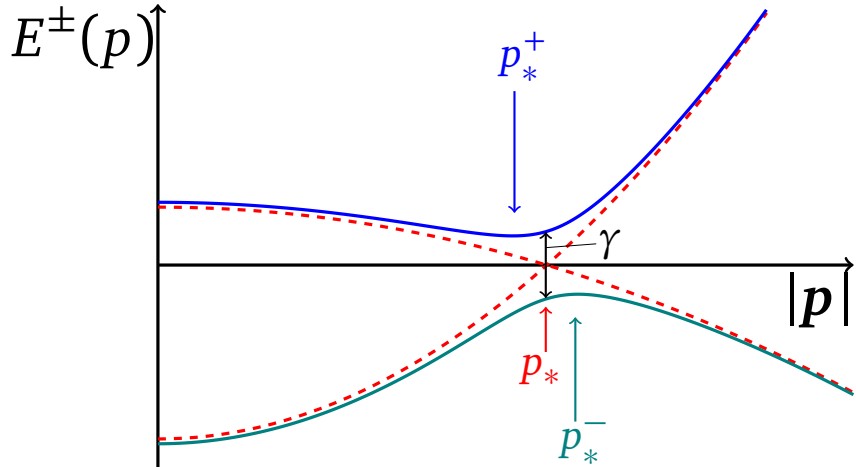

Figure 1: Overview of insulating bandstructure at zero field. The unhybridised bands (dashed lines) cross at momentum $|\boldsymbol{p}| = p_*$. The hybridisation gap $\gamma$ leads to two bands, $E^\pm(p)$, which have their minimum / maximum at $p_*^\pm$.

Consider now the application of a perpendicular magnetic field $B$, which couples to the orbital motion of the electrons. Landau quantization leads to the replacements $|\hat{\boldsymbol{p}}|^2/(2m_i) \to \hbar\omega_i(n + 1/2)$ for each band $i = 1, 2$, where $n = 0, 1, 2 \ldots$ is the Landau level index and $\omega_i = eB/m_i$ are the cyclotron frequencies. Since the hybridisation is spatially independent, the Hamiltonian is diagonal in the Landau level index

$$H_n = \begin{pmatrix} \hbar\omega_1(n - n_*) & \gamma/2 \\ \gamma/2 & -\hbar\omega_2(n - n_*) \end{pmatrix}, \tag{3}$$

where

$$n_* = \frac{p_*^2}{2eB\hbar} - 1/2. \tag{4}$$

Each state has a degeneracy of $n_\phi A$ where $n_\phi = eB/h$ is the flux density and $A$ is the total area of the 2D system. The energy eigenvalues are

$$E_n^\pm = \frac{1}{2}\left[ \hbar(\omega_1 - \omega_2)(n - n_*) \pm \sqrt{[\hbar(\omega_1 + \omega_2)(n - n_*)]^2 + \gamma^2} \right], \tag{5}$$

and we denote the eigenstates by

$$|\Psi_n^+\rangle = \begin{pmatrix} \alpha_n \\ \beta_n \end{pmatrix}, \qquad |\Psi_n^-\rangle = \begin{pmatrix} -\beta_n \\ \alpha_n \end{pmatrix}, \tag{6}$$

with $\alpha_n = \gamma/\sqrt{\gamma^2 + (2E_n^-)^2}$ and $\beta_n = \sqrt{1 - \alpha_n^2}$.

In the absence of hybridization, $\gamma = 0$, the system is a metal. In a magnetic field, the value $n_*$ controls the precise properties of this metal close to the band touching point. For integer $n_*$ the two bands touch, with an exact degeneracy of the Landau level at $n = n_*$. For non-integer $n_*$, there is a residual gap between the two bands of magnitude $\lesssim \hbar(\omega_1 + \omega_2)$. This gap opens and closes each time that $n_*$ changes by 1, that is with fundamental period

$$\Delta(1/B) = \frac{2e\hbar}{p_*^2} = \frac{2\pi e}{\hbar S_*}, \tag{7}$$

where $S_* \equiv \pi(p_*/\hbar)^2$ is the area traced out by the gap-closing point in reciprocal space. Concomitant with the gap oscillation, at temperatures that are not much larger than $\hbar(\omega_1 + \omega_2)$, there is an oscillation in the total energy of the system (given by the occupied energy levels). This oscillation of the total energy gives rise to the dHvA effect of the metal [1].

For a non-zero hybridisation $\gamma$, the system is an insulator. There is then always a gap close to $n_*$ for any value of the magnetic field. However, provided the hybridisation gap $\gamma$ remains small compared to the oscillations of the gap in the metal, i.e.

$$\hbar(\omega_1 + \omega_2) \gtrsim \gamma, \tag{8}$$

then the total energy of the occupied states of the insulator will oscillate in a similar manner to the total energy of the metal, so one expects there still to be a dHvA effect in the insulator. In essence this is the content of the theories for quantum oscillations of the magnetization in band insulators [20–22, 33]. A full calculation for this model shows that the oscillatory part of the energy is [23]

$$E_{\text{osc}}^{\text{BI}} = \frac{\sqrt{\gamma \hbar(\omega_1 + \omega_2)}}{2} n_\phi \sum_{k>0} \frac{\cos(2\pi k n_*)}{k^{3/2}} \exp\left[-\frac{2\pi \gamma k}{\hbar(\omega_1 + \omega_2)}\right], \tag{9}$$

where the integer $k$ defines the harmonic of the fundamental period (7). The suppression as $B \to 0$ resembles the functional form of "Dingle damping" of quantum oscillations in a metal due to impurity scattering, with $E_{\text{osc}}^{\text{BI}} \sim \exp(-B_0^{\text{BI}}/B)$. Here the fundamental oscillation period ($k = 1$) sets the characteristic field scale

$$B_0^{\text{BI}} = \frac{2\pi}{(m_1^{-1} + m_2^{-1})} \frac{\gamma}{e\hbar}. \tag{10}$$

For the band insulator this suppression is not due to impurities, but is an intrinsic effect that is tied to the nonzero hybridization gap. Note that the manner in which the hybridization gap $\gamma$ determines the non-zero $B_0^{\text{BI}}$ is very much model-dependent. Eqn. (10) is valid for the model studied here (of two parabolic bands). Other models – with three bands, or with non-parabolic band dispersion – can give very different sizes of $B_0^{\text{BI}}$ for the same $\gamma$ [54].

## 3 Impurity states

To model impurity states in a simple manner, we introduce an attractive delta-function potential of strength $V_0$

$$\hat{V} = -V_0 \delta^2(\boldsymbol{r}) \mathbb{1}, \tag{11}$$

where the $\mathbb{1}$ refers to the matrix structure in (1) such that the potential acts equivalently on the two bands. This potential should be viewed as a caricature of the potential of an ionized donor impurity atom, which is expected to be largely Coulombic though with short-distance corrections [55].

A full solution of the problem of boundstates of the Landau quantized levels (3) on the contact potential (11) is provided in Appendix A. Here we focus on shallow bound states which lie just below the band edge of the upper band, $E_n^+$, with a binding energy $E_{\text{B}}$ that is small compared to the hybridzation gap $\gamma$. To describe these, it is sufficient to consider the energy levels close to the energy minimum of $E_n^+$, which is located at

$$n_*^+ = \frac{(p_*^+)^2}{2eB\hbar} - \frac{1}{2}, \tag{12}$$

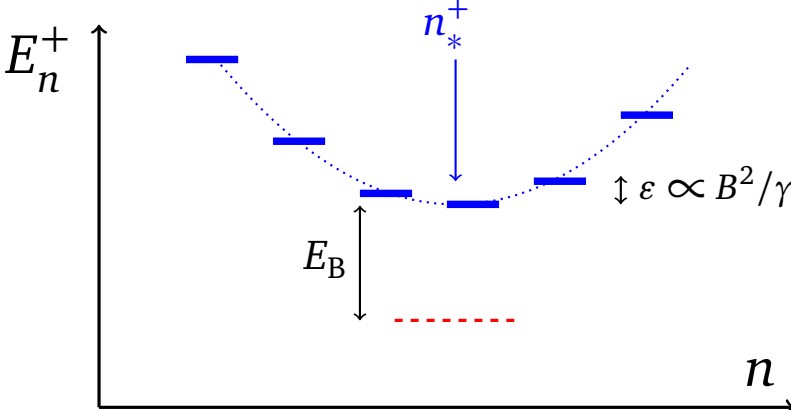

Figure 2: Close-up of the Landau-quantized energy levels close to the conduction band minimum in the parabolic approximation (13). The characteristic energy spacing is set by $\varepsilon$, Eqn (15). As the magnetic field varies, the Landau levels pass through the minimum $n_*^+$ with fundamental period $\Delta(1/B) = 2\pi e/(\hbar S_*^+)$. An impurity state, with binding energy $E_{\mathrm{B}}$, is denoted by the dashed line.

and close to which the energies can be approximated by

$$E_n^+ \simeq E_*^+ + \frac{1}{2}\varepsilon(n - n_*^+)^2, \tag{13}$$

$$E_*^+ \equiv \frac{\gamma\sqrt{m_1 m_2}}{(m_1 + m_2)}, \tag{14}$$

$$\varepsilon \equiv \frac{4\hbar^2(\omega_1\omega_2)^{3/2}}{\gamma(\omega_1 + \omega_2)}. \tag{15}$$

Using this quadratic form (13) and taking the wavefunctions $\alpha_n, \beta_n$ to be independent of $n$, the condition for a state of energy $E = E_*^+ - E_{\mathrm{B}}$ (i.e. with binding energy $E_{\mathrm{B}}$ below the minimum energy $E_*^+$ of the upper band) is

$$\frac{1}{V_0} = \frac{1}{2\pi\ell_B^2} \sum_{n=-\infty}^{\infty} \frac{1}{E_{\mathrm{B}} + (\varepsilon/2)(n - n_*^+)^2}, \tag{16}$$

where $\ell_B = \sqrt{\hbar/eB}$ is the magnetic length. Note that the limit of the sum has been extended, consistent with the dominance of terms close to $n_*^+$. In fact, the full theory for the delta-function potential (11) described in Appendix A requires a short-distance cut-off, such as the lattice constant. In the regimes we consider, of a shallow boundstate, the cut-off dependence is small and can be safely ignored for the quantities of primary interest here.

The analysis of equation (16) leads to the key results of this paper.

## 3.1 Bound state for $B = 0$

Consider first the case of vanishing magnetic field, for which we denote the binding energy of the impurity level by $E_{\mathrm{B}}^0$. In the limit $B \to 0$ the sum over $n$ can be replaced by an integral and Eqn (16) becomes

$$\frac{1}{V_0} = \frac{1}{\ell_B^2}\sqrt{\frac{1}{2\varepsilon E_{\mathrm{B}}^0}}. \tag{17}$$

The magnetic field $B$ drops out to leave

$$E_{\text{B}}^0 = \frac{V_0^2 \gamma}{8\hbar^4} (m_1 m_2)^{1/2} (m_1 + m_2). \tag{18}$$

The corresponding boundstate wavefunction has an interesting spatial structure which combines features of one and two dimensions [46, 56]. The one-dimensional features arise from the fact that the dispersion $E^+(\boldsymbol{p})$ has a ring-like minimum at $|\boldsymbol{p}| = p_*^+$, so depends only on the radial momentum as $E^+(\boldsymbol{p}) \simeq E_*^+ + (|\boldsymbol{p}| - p_*^+)^2/(2m_*^+)$ giving a one-dimensional density of states at the band edge, characterized by the effective mass

$$m_*^+ = \gamma \sqrt{m_1 m_2} (m_1 + m_2)/(2p_*^+)^2. \tag{19}$$

The bound state wavefunction in momentum space takes the form $\psi_{\boldsymbol{p}} \propto [1 + a_0^2 (|\boldsymbol{p}| - p_*^+)^2/\hbar^2)]^{-1}$, with the lengthscale $a_0 = \hbar/\sqrt{2m_*^+ E_{\text{B}}^0}$. The overall 2D spatial wavefunction takes the form [46]

$$\psi(\boldsymbol{r}) \simeq J_0(p_*^+ r/\hbar) \exp(-r/a_0), \tag{20}$$

which combines exponential decay on the scale $a_0$ with oscillations on the scale $\lambda_*^+ \equiv 2\pi\hbar/p_*^+$, the characteristic wavelength at the band minimum. Since we are considering regimes of small binding energy $E_{\text{B}}^0 \ll (p_*^+)^2/(2m_*^+)$, this wavelength is small compared to the boundstate size, $\lambda_*^+ \ll a_0$, giving a highly oscillatory boundstate wavefunction.

## 3.2 Quantum oscillations of the bound state, $B \neq 0$

For non-zero $B$ there are corrections to the energy of the impurity level that oscillate with inverse magnetic field. In the limit that the binding energy goes to zero, the energy of the impurity level is tied to that of the band edge, so these corrections to the energy will simply reflect the oscillations of the free-particle states near the band edge (13). However, for nonzero binding energy, the oscillations are reduced. We compute the corrections to the energy of the impurity for non-zero binding energy using the Poisson summation formula, writing Eq. (16) as

$$\frac{2\pi \ell_B^2}{V_0} = \int_{-\infty}^{\infty} \mathrm{d}n f(n - n_*^+) + I_{\text{osc}}(n_*^+), \tag{21}$$

where $f(n) \equiv 1/[E_{\text{B}} + (\varepsilon/2)n^2]$, and the part that oscillates with $n_*^+$ is

$$I_{\text{osc}}(n_*^+) = \sum_{k \neq 0} \mathrm{e}^{-\mathrm{i}2\pi k n_*^+} \int_{-\infty}^{\infty} \mathrm{d}n' f(n') \mathrm{e}^{-\mathrm{i}2\pi k n'}, \tag{22}$$

with $k$ integer. We treat $I_{\text{osc}}$ as small, and write the binding energy as $E_{\text{B}} = E_{\text{B}}^0 - E_{\text{osc}}^{\text{imp}}$ where $E_{\text{B}}^0$ is the $B = 0$ binding energy and $E_{\text{osc}}^{\text{imp}}$ is the $B$-dependent correction to the energy of the impurity level. Computing $E_{\text{osc}}^{\text{imp}}$ to first order in $I_{\text{osc}}$, and using (17), the condition (21) becomes

$$E_{\text{osc}}^{\text{imp}} = -4E_{\text{B}}^0 \sum_{k>0} \cos(2\pi k n_*^+) \exp\left(-2\pi|k| \sqrt{2E_{\text{B}}^0/\varepsilon}\right). \tag{23}$$

Reintroducing the expression for the energy scale $\varepsilon$ (15) leads to

$$E_{\text{osc}}^{\text{imp}} = -4E_{\text{B}}^0 \sum_{k>0} \cos(2\pi k n_*^+) \exp\left[-\frac{\pi|k|(\omega_1 + \omega_2)^{1/2}\sqrt{2\gamma E_{\text{B}}^0}}{\hbar(\omega_1 \omega_2)^{3/4}}\right]. \tag{24}$$

Equation (24) is a key result of this work. It shows that the localised states are subject to quantum oscillations in their energies.

These oscillations have several notable features:

(i) The oscillatory contributions arise as $n_*^+$, Eqn (12), increases by an integer divided by $k$. This leads to periods in $1/B$ of

$$\Delta(1/B) = \frac{1}{k}\frac{2\pi e}{\hbar S_*^+}, \qquad (25)$$

where $S_*^+ \equiv \pi(p_*^+/\hbar)^2$ is the area in reciprocal space associated with the band minimum of the upper band, $E_*^+$. The oscillatory energy of the bound state reflects the oscillatory structure of the quantized energy levels in the vicinity of the band extremum. Had we focused on states close to the top of the lower band, the relevant area would have been $S_*^- \equiv \pi(p_*^-/\hbar)^2$.

(ii) The oscillations of the impurity energy level are exponentially suppressed at small $B$ through the dependence of the form $\exp(-B_0^{\text{imp}}/B)$, with the characteristic field scale

$$B_0^{\text{imp}} = \pi(m_1 m_2)^{1/4}(m_1 + m_2)^{1/2}\frac{\sqrt{2\gamma E_{\text{B}}^0}}{e\hbar}, \qquad (26)$$

set by the fundamental oscillation period ($k = 1$). This has a rather different form from the field scale for quantum oscillations of the band insulator (10). Consequently, the suppression can be larger or smaller than that in the band insulator (10), depending on details of parameters. For $m_1 = m_2$, we have $B_0^{\text{imp}}/B_0^{\text{BI}} = 2\sqrt{E_{\text{B}}^0/\gamma}$ which is typically small for the shallow bound states we consider, $E_{\text{B}}^0 \ll \gamma$. Then, the oscillations of the energy of the boundstate will persist to lower fields than those of the energy of the bulk insulator.

A simple understanding of the characteristic magnetic field scale (26) can be obtained by a comparison of the binding energy $E_{\text{B}}^0$ with the energy dispersion of the upper band, $E_n^+$, in the vicinity of its minimum. The dominant contributions to the bound state wavefunction come from free-particle states of energies up to $\sim E_{\text{B}}^0$ above this energy minimum. Within the quadratic approximation (13), these free-particle states span a number of Landau levels $(\Delta n) \sim \sqrt{E_{\text{B}}/\varepsilon}$. When this number is large $\Delta n \gg 1$, one expects a weak dependence on the magnetic field. Conversely, when this number is of order unity, $\Delta n \lesssim 1$, one expects the bound state to depend strongly on magnetic field, acquiring the periodic oscillations of the single particle energies at this band minimum. The condition $\Delta n \lesssim 1$ leads to the condition for visibility of the oscillations to be $B \gtrsim B_0^{\text{imp}}$ with $B_0^{\text{imp}}$ given by (26) up to an overall numerical factor. While the detailed calculation leading to (26) was performed for a contact potential (11), the reasoning just given indicates that the same general scaling, giving the condition $\sqrt{E_{\text{B}}^0/\varepsilon} \lesssim 1$ for visibility of oscillations, should hold also for more general potentials.

Recalling that the boundstate wavefunction is characterised by its overall size $a_0$ and short-range modulations on the scale $\lambda_*^+$ (the wavelength at the band minimum), the field-dependent damping factor determining the visibility of the quantum oscillations of the bound-state energy, $\exp(-B_0^{\text{imp}}/B)$, can also be written

$$\exp\left(-\frac{2\pi}{n_\phi a_0 \lambda_*^+}\right). \qquad (27)$$

Thus, of order one flux quantum (or more) should thread through an area $a_0 \lambda_*^+$ to effect sizeable oscillations of the energy of the impurity level. The bound state wavefunction (20) is rotationally symmetric, with an overall extent $a_0$ and with an amplitude that alternates in sign on a length scale of $\lambda_*^+/2$. The area $a_0 \lambda_*^+$ can be viewed as the typical area over which the wavefunction is of constant sign: a ring of circumference $\sim a_0$ and width $\sim \lambda_*^+$. Perhaps more helpful is to interpret the condition in terms of the free-particle states. Within

the semiclassical description of the motion of the electron in a magnetic field, the typical spatial extent of the unbound Landau-quantized state in the vicinity of the energy minimum $p_*^+$ is set by the cyclotron radius $R_c = \ell_B^2 p_*^+/\hbar = 2\pi\ell_B^2/\lambda_*^+$. Using this, the field-dependent damping factor (27) can therefore also be written

$$\exp\left(-\frac{2\pi R_c}{a_0}\right). \tag{28}$$

This gives the very intuitive criterion that the quantum oscillations will be sizeable provided the cyclotron orbit fits within the overall spatial extent of the boundstate, $R_c \lesssim a_0$. It can also be helpful to interpret how this condition arises in reciprocal space. In a disorder-free 2D system with a perpendicular magnetic field $B$, the semiclassical orbits in momentum space are closed loops with areas that increase by $2\pi\hbar^2/\ell_B^2$ from one state to the next [19]. Here the relevant states (close to the band edge) are circles of radius close to $p_*^+$, so their radii increase in steps of $\Delta p$ set by $(\Delta p)2\pi p_*^+ = 2\pi\hbar^2/\ell_B^2$, i.e. $\Delta p = \hbar/R_c$. Localisation to a lengthscale $a_0$ leads to an uncertainty of radial momentum of $\sim \hbar/a_0$. Thus, for $\hbar/a_0 \gtrsim \hbar/R_c$ the semiclassical quantization condition is washed out, and one expects suppression of the quantum oscillations, which reproduces the same criterion.

## 4 Experimental consequences

We have studied the properties of a single impurity level, showing that it exhibits quantum oscillations that are inherited from the structure of the underlying energy bands. For a system with a small nonzero density of such impurity levels, $n_{\text{imp}}$, with random locations, one expects that the many-electron groundstate will be an Anderson insulator. The groundstate will have electrons bound to the impurity sites, with wavefunctions that decay exponentially in space. In this regime, the zero temperature limit of the conductivity will vanish. However, given that each impurity level oscillates according to (23), one expects an oscillation of the energy per unit area of the impurity band of $E_{\text{osc}}^{\text{imp}-\text{B}} = n_{\text{imp}}E_{\text{osc}}^{\text{imp}}$. For sufficiently dilute impurities the inter-site hybridisation will be small compared to the binding energy of the impurity level, and this oscillation of the single-impurity level will dominate the energetics of the electrons in the impurity band. (As we discuss below, for the model we study the hybridisation between impurity levels is expected to be strongly suppressed even for overlapping impurities [55].) Taking (minus) the derivative of this with respect to magnetic field gives oscillations in the magnetization per unit area of magnitude

$$M_{\text{osc}}^{\text{imp}-\text{B}} = \frac{4\pi E_{\text{B}}^0 (p_*^+)^2}{e\hbar B^2} n_{\text{imp}} \exp\left(-B_0^{\text{imp}}/B\right), \tag{29}$$

where we have taken the fundamental period ($k = 1$). We note in passing that the same argument leads to the expectation that the energetics of a Mott insulator formed in this impurity band[1] will be dominated by the impurity energy level, and that the Mott insulator will also exhibit a dHvA effect given by (29).

By comparison, using (9) the magnitude of the $k = 1$ oscillations in the magnetization per unit area due to the background band insulator is

$$M_{\text{osc}}^{\text{BI}} = \frac{\pi p_*^2 \sqrt{\gamma\hbar(\omega_1 + \omega_2)}}{2eB^2\hbar} n_\phi \exp\left(-B_0^{\text{BI}}/B\right). \tag{30}$$

---

[1]A Mott insulator would arise in our model if we were to allow the impurity levels to host two spin states, and consider a half-filled impurity band with strong short-range electron-electron repulsion.

In the Anderson insulator there are contributions from *both* the impurity band (29) and from the filled band insulator (30). Depending on parameters, either one of these contributions can dominate. However, for mass ratios $m_2/m_1 \sim 1$ and shallow bound states $E_{\mathrm{B}}^0 \lesssim \gamma$ the contributions from the impurity band are typically the larger at small fields, $B \to 0$.

Similar energetic modulations of an impurity band are captured by theories which model the effects of impurities through phenomenological scattering rates for the bands [32, 33]. This also leads to a Dingle-damping-like suppression of oscillations at small $B$, with a factor $B_0$ now controlled by these scattering rates. However, such theories, which model the impurity band by a finite electron lifetime, cannot capture Anderson localisation, since they lead to a non-zero value of the conductivity even in the limit of zero temperature [33].

For very small impurity concentration, and in an otherwise pristine band insulator, the localized electronic levels in the impurity band are separated by a clean gap $E_{\mathrm{B}}^0$ from the extended states of the $E^+$ band. One therefore expects that, for small non-zero temperatures, $k_B T \ll E_{\mathrm{B}}^0$, the electron occupation of the impurity band will be reduced in an activated form. However, the thermally excited electrons can themselves show quantum oscillations, owing to the nonzero spacing $\varepsilon$ separating the Landau levels in the upper band [33, 34]. At the lowest temperatures these thermally excited electrons add to the oscillation amplitude, before becoming suppressed for $k_B T \gg \varepsilon$. Fig. 3 provides an illustration of the temperature dependence of the magnetization arising from the impurity band and thermal excitation into unbound levels. (See Appendix B for more details.) As a more realistic model of the impurity band, suitable for impurity states that are strongly hybridised or subject to long-range Coulomb interactions, one should introduce a nonzero width to the impurity band, and thus average over a range of binding energies $E_0^B$. A large broadening of the impurity band can remove the gap in the density of states between the impurity band and the continuum of unbound states in the upper band, without destroying Anderson localization. In this case, the dHvA effect of the unbound states may persist down to zero temperature, removing the dip seen in Fig. 3 at low temperatures. In effect, this would be similar to pinning the chemical potential in the upper band, which can lead to a low-temperature enhancement of the dHvA effect coming from the unbound states [20].

At large impurity density $n_{\mathrm{imp}}$ the wavefunctions on different impurity levels will overlap and lead to hybridisation of the electronic wavefunctions. Since we study a system in two dimensions, for a disordered configuration of impurities, this hybridisation will not lead to delocalization: the electrons in the impurity band will remain Anderson-localized [57]. In three dimensional settings, there can be a transition to a delocalized phase. In conventional materials, with a simple band minimum at one point in the Brillouin zone and accounting for the fact that electrons interact through Coulomb interactions, the transition to the metal occurs at the Mott criterion [58] $n_{\mathrm{imp}}^{\mathrm{3D}} \gtrsim 0.26/a_0^3$ where $a_0$ is the characteristic size of the localized level. However, Skinner [55] has argued that for materials of the form we study, in which the band minimum is on a surface $|\boldsymbol{p}| = p^*$, the insulating state can remain robust despite an apparent violation of the Mott criterion. This form of band structure leads to the additional oscillatory structure of the hydrogenic bound states, as in (20) for 2D, which significantly reduces the hybridisation of the impurity levels: the hybridisation remains small compared to the single-impurity binding energy, and the system can remain an Anderson insulator, even for strongly overlapping orbitals, up to $n_{\mathrm{imp}}^{\mathrm{3D}} \simeq (p_*/\hbar)^3$. The results that we present here show that this special form of the bandstructure *also* guarantees the existence of quantum oscillations of the magnetization in such an Anderson insulator. For overlapping impurities the localization length of the electronic states can be extend beyond $a_0$. In this regime, the form of the field damping will not necessarily follow that of the dilute impurity case that we have presented. Rather, drawing on the discussions following (28), we expect that the quantum oscillations will be suppressed when the free-particle cyclotron radius $R_c = \ell_B^2 p_*^+/\hbar$ is larger than the mean free

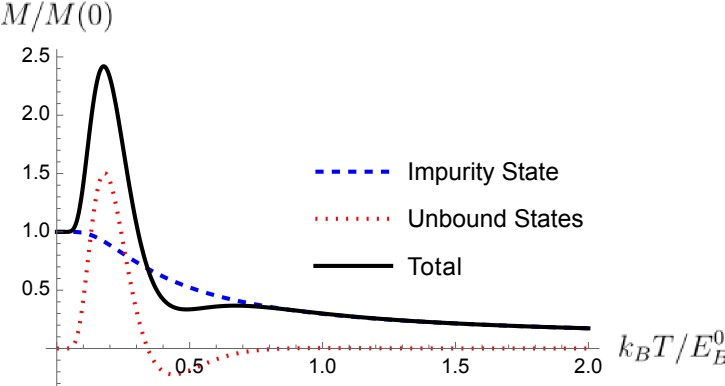

Figure 3: Temperature dependence of the impurity band contribution to the dHvA effect in an Anderson insulator, showing the amplitude of the oscillation at the fundamental period, $\Delta(1/B) = 2\pi e/(\hbar S_*^+)$. The parameters are $n_\phi = 2n_{\text{imp}}$ and $\varepsilon = E_{\text{B}}^0$, and the chemical potential is varied to keep the electron number fixed at $n_{\text{imp}}$. (See Appendix B for details.) At zero temperature all electrons are bound to impurities and the oscillation amplitude of the magnetization is $M(0) = M_{\text{osc}}^{\text{imp}-\text{B}}$, Eqn (29). At non-zero temperatures, electrons are thermally excited to unbound states in the upper band. The oscillation amplitude is at first enhanced by thermal excitation, as the states in the upper band also contribute to the dHvA effect via the quantum oscillations of their energies (13). The contributions of the unbound states are suppressed for $k_B T \gg \varepsilon$, and the impurity is eventually unpopulated for $k_B T \gg E_{\text{B}}^0$.

path $\ell_{\text{mfp}}$ which one can interpret as setting the uncertainty in the momentum of the electron to $\sim \hbar/\ell_{\text{mfp}}$. That is, the oscillations are suppressed for $R_c \gtrsim \ell_{\text{mfp}}$ in place of $R_c \gtrsim a_0$ from (28), increasing the field range under which they can contribute if $\ell_{\text{mfp}} > a_0$. Note that the mean free path is shorter than the localization length, and indeed it typically remains finite even in a metallic phase where the localization length diverges. In that regime, the condition $R_c \gtrsim \ell_{\text{mfp}}$ will account for the disorder damping of quantum oscillations in this metallic phase. A full calculation of the dependence of the mean free path on impurity concentration for the model of a non-parabolic band with an energy minimum at $p_*^+$ is beyond the scope of the present paper.

One of the conclusions of Ref. [55] was that a model of an impurity-band insulator, formed from a density $n_{\text{imp}}^{\text{3D}}$ of impurities, could account for the anomalously large low-temperature heat capacities of the Kondo insulators $SmB_6$ and $YbB_{12}$. These heat capacities have been measured to be linear in $T$ at low temperatures, rather than having the activated form expected for pristine band insulators. Our results show that this Anderson insulator will also exhibit oscillations of the magnetisation, and allow us to estimate the sizes of the contributions to this dHvA effect. It is difficult to make accurate quantitative comparisons between our theory and experiment given uncertainties in the microscopic parameters for the Kondo insulators. The model studied here is also likely too simple to capture all quantitative features of these strongly interacting materials. That said, proceeding cautiously, we note that the mean-field theory for a Kondo insulator leads to a theory of the form that we have studied, with the band masses $m_1$ and $m_2$ arising from the $d$ and $f$ bands. Thus, the mass ratio is expected to be very large $m_2/m_1 \gtrsim 30$, and consequently $B_0^{\text{imp}}$ is typically large compared to $B_0^{\text{BI}}$. This comparison suggests that, for the parameters of the impurity-band insulator discussed in Ref. [55], the $T = 0$ dHvA effect at weak magnetic fields would likely be dominated by the contributions from the band insulator (30) with the contributions from the impurity band (29) being more strongly suppressed since $B_0^{\text{imp}} > B_0^{\text{BI}}$. Note that recent work [41] has shown that for insulators

in which the hybridization $\gamma$ is driven by interactions (as for Kondo insulators) there can be very significant contributions to the dHvA effect of the band insulator from beyond-mean-field effects associated with quantum fluctuations of the gap, which can very greatly enhance the size of the effect (9). In the situation where the $T = 0$ dHvA effect at weak magnetic fields is dominated by the contributions from the band insulator, a theory that takes these quantum fluctuations into account will surely be required to give a suitable quantitative estimate of size of the dHvA effect in Kondo insulators.[2]

Further exploration of the role of the impurity band beyond what we have studied in this paper would be worthwhile. We recall that we expect significant changes to our results if the impurity band is sufficiently broadened to close the gap in the density of states. Furthermore, the bulk parameters we have used above [55] do not take into account any possible suppression of the hybridisation gap in the vicinity of the impurity, which could significantly change the quantitative description of the impurity boundstate [59]. For now, using the estimates from Ref. [59] of cyclotron radius $R_c \simeq 100$nm and of the size of the boundstates around impurities, $\sim 3$nm, our result (28) would predict a significant suppression of the quantum oscillations from the individual impurity levels.

## 5    Summary and Outlook

We have shown that an Anderson insulator can give rise to quantum oscillations of the magnetization. This occurs in situations where the electronic band from which the localized states are constructed has a special form: with a band minimum that traces out a closed area in reciprocal space. The quantum oscillations of the impurity levels are controlled by this characteristic area in reciprocal space, $S_*^+ = \pi(p_*^+/\hbar)^2$, even though they arise in a disordered setting where the contributing electronic states are strongly localized, and therefore are unable to conduct. This area remains physically relevant provided the size of the bound state – i.e. the localization length $a_0$ – is large compared to the characteristic wavelength $\lambda_*^+ \equiv 2\pi\hbar/p_*^+$ of the underlying bandstructure. Our results show the appearance of quantum oscillations of the impurity boundstate energy when the cyclotron radius $R_c = 2\pi\ell_B^2/\lambda_*^+$ fits within the size of the boundstate $a_0$.

The contributions of the impurity levels to the dHvA effect is damped exponentially at weak field, with a characteristic field (26) that differs from that for the band insulator (10). The dHvA effect in an Anderson insulator has contributions from both, and either one or the other can dominate depending on parameters. We discussed the temperature dependence of the oscillation, which can lead to a non-monotonic dependence. Making a comparison to models for the Kondo insulators, we found that use of the bulk parameters in the simple two-band model studied here is likely to preclude the relevance of a narrow impurity band to the dHvA effect.

To observe a dHvA effect in an Anderson insulator that is dominated by the impurity band contribution it is most useful to study experimental systems where the mass ratio $m_2/m_1$ is of order one. Suitable systems naturally arise in band insulators of InAs/GaSb quantum well structures [29], or in bilayer graphene in a perpendicular electric field [46,60]. These systems have light masses, allowing large cyclotron frequencies at moderate magnetic fields, and the band overlap (setting $p_*$) and the hybridisation ($\gamma$) can readily be tuned experimentally. Oscillations of the in-gap conductance have been reported in InAs/GaSb systems [10–12], but so far we are unaware of experimental measurements of the oscillation of the magnetisation. Our results show that it can take a rich form, with distinct contributions arising both the bulk bands of the pristine insulators and from impurity levels.

---

[2]From the nature of that theory [41] we do not expect similar corrections to the impurity levels.

# Acknowledgements

We are grateful to Andrew Allocca, Johannes Knolle and Suchitra Sebastian for their expert insights on this topic communicated through long-standing discussions, and for helpful comments on the manuscript.

**Funding information** This work was supported by EPSRC Grant No EP/P034616/1 and by a Simons Investigator Award, number 511029.

# A  General bound state condition

To study the bound state of the contact interaction (11) in the presence of a magnetic field it is helpful to consider the position representation of the Landau level states. We use the symmetric gauge for which the angular momentum $m$ is a good quantum number. The contact interaction acts only on states with angular momentum $m = 0$, for which the Landau level wavefunctions may be taken to be

$$\psi_{n,m=0}(\boldsymbol{r}) = \frac{1}{\sqrt{2\pi\ell_B^2}} e^{-r^2/4\ell_B^2} L_n(r^2/2\ell_B^2)\,, \tag{A.1}$$

with $L_n(z)$ the Laguerre polynomials and $\ell_B = \sqrt{\hbar/eB}$ the magnetic length. We expand the Schrödinger equation for a state of energy $E$ in the presence of the delta-function potential in this basis, as

$$|\Psi\rangle = \sum_n \phi_n^+ |\Psi_n^+\rangle + \phi_n^- |\Psi_n^-\rangle\,. \tag{A.2}$$

The relevant matrix elements of the contact interaction (11) are

$$\begin{pmatrix} \langle\Psi_{n'}^+|\hat{V}|\Psi_n^+\rangle & \langle\Psi_{n'}^+|\hat{V}|\Psi_n^-\rangle \\ \langle\Psi_{n'}^-|\hat{V}|\Psi_n^+\rangle & \langle\Psi_{n'}^-|\hat{V}|\Psi_n^-\rangle \end{pmatrix} = -\frac{V_0}{2\pi\ell_B^2} \begin{pmatrix} \alpha_{n'}\alpha_n + \beta_{n'}\beta_n & -\alpha_{n'}\beta_n + \beta_{n'}\alpha_n \\ -\beta_{n'}\alpha_n + \alpha_{n'}\beta_n & \alpha_{n'}\alpha_n + \beta_{n'}\beta_n \end{pmatrix}\,. \tag{A.3}$$

Defining the quantities $M^\pm \equiv \sum_n \phi_n^\pm \alpha_n$ and $N^\pm \equiv \sum_n \phi_n^\pm \beta_n$ one finds

$$M^\pm = -\frac{V_0}{2\pi\ell_B^2}\left(S_{\alpha\alpha}^\pm M^\pm + S_{\alpha\beta}^\pm N^\pm \mp S_{\alpha\beta}^\pm M^\mp \pm S_{\alpha\alpha}^\pm N^\mp\right)\,, \tag{A.4}$$

$$N^\pm = -\frac{V_0}{2\pi\ell_B^2}\left(S_{\alpha\beta}^\pm M^\pm + S_{\beta\beta}^\pm N^\pm \mp S_{\beta\beta}^\pm M^\mp \pm S_{\alpha\beta}^\pm N^\mp\right)\,, \tag{A.5}$$

where

$$S_{\alpha\alpha}^\pm(E) = \sum_{n=0}^{n_{\max}} \frac{\alpha_n^2}{E - E_n^\pm}\,, \quad S_{\alpha\beta}^\pm(E) = \sum_{n=0}^{n_{\max}} \frac{\alpha_n\beta_n}{E - E_n^\pm}\,, \quad S_{\beta\beta}^\pm(E) = \sum_{n=0}^{n_{\max}} \frac{\beta_n^2}{E - E_n^\pm}\,. \tag{A.6}$$

Hence, for a normalizable eigenstate at energy $E$, one requires

$$\det \begin{pmatrix} S_{\alpha\alpha}^+(E) + \frac{2\pi\ell_B^2}{V_0} & S_{\alpha\beta}^+(E) & -S_{\alpha\beta}^+(E) & S_{\alpha\alpha}^+(E) \\ S_{\alpha\beta}^+(E) & S_{\beta\beta}^+(E) + \frac{2\pi\ell_B^2}{V_0} & -S_{\beta\beta}^+(E) & S_{\alpha\beta}^+(E) \\ S_{\alpha\beta}^-(E) & -S_{\alpha\alpha}^-(E) & S_{\alpha\alpha}^-(E) + \frac{2\pi\ell_B^2}{V_0} & S_{\alpha\beta}^-(E) \\ S_{\beta\beta}^-(E) & -S_{\alpha\beta}^-(E) & S_{\alpha\beta}^-(E) & S_{\beta\beta}^-(E) + \frac{2\pi\ell_B^2}{V_0} \end{pmatrix} = 0\,. \tag{A.7}$$

We have introduced a limit $n_{\max}$ in the sums in (A.6) to regularize a divergence that arises for the pure delta-function potential. For large $n$, using the forms of the energies $E_n^\pm$ and

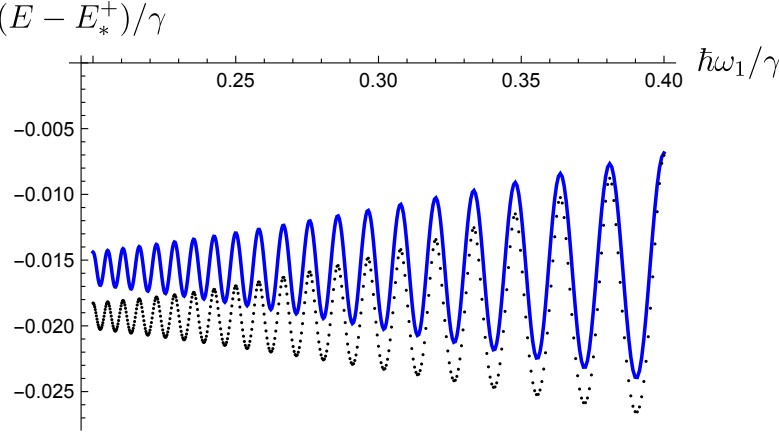

Figure 4: Quantum oscillations of the energy of an impurity level, close to the upper band, with magnetic field, $\hbar\omega_1/\gamma \propto B$. Black points: Exact result from the numerical solution of (A.7). Solid blue line: analytic result obtained from the analysis of (A.8), which leads to $E = E_*^+ - E_B^0 + E_{\text{osc}}^{\text{imp}}$ from Eqns. (14, 18, 23). The discrepancy reflects the correction introduced to the binding energy from the high-energy part of the spectrum, which is ignored in deriving (16). The parameters used are $m_1 = m_2$, $V_0 m_1/\hbar^2 = 0.25$, $n_{\text{max}}\hbar\omega_1 = 20\gamma$ and $p_*^2/(2m_1) = 8\gamma$.

wavefunction coefficients $\alpha_n$ and $\beta_n$, one finds that $S_{\beta\beta}^\pm$ acquire logarithmic divergences with the maximum Landau level index included, i.e. $S_{\beta\beta}^\pm \sim \mp[1/(\hbar\omega_{1,2})]\log n_{\text{max}}$. This cut-off sets a minimum lengthscale for the potential, $\sim \ell_B/\sqrt{n_{\text{max}}}$, which physically is set by the microscopics of the system. For example a natural scale is the lattice constant $\lambda_0$, in terms of which we take $n_{\text{max}} = \ell_B^2/\lambda_0^2$.

Eqn.(A.7) may be efficiently solved numerically to find the in-gap boundstate and its dependence on magnetic field. In Fig. 4 we show the magnetic field dependence of a bound state close to the upper band.

In the main text we study a shallow boundstate close to the minimum of $E_n^+$, i.e. with Landau level index $n$ close to $n_*^+$. Then we may restrict attention to the functions $S_{\mu\nu}^+(E)$, use the quadratic expansion (13) and take $\alpha_n \simeq \alpha_{n_*^+}$ and $\beta_n \simeq \beta_{n_*^+}$. The functions $S_{\mu\nu}^+(E)$ then differ only by prefactors and the condition (A.7) reduces to

$$\frac{2\pi\ell_B^2}{V_0} = \sum_{n=-\infty}^{\infty} \frac{1}{E_B + (\varepsilon/2)(n - n_*^+)^2}, \tag{A.8}$$

where we have introduced the binding energy $E_B \equiv E_*^+ - E$. Note that the limits of the sum have been extended consistent with the dominance of terms close to $n = n_*^+$.

The effect of the cut-off $n_{\text{max}}$ can be judged by using the exact dispersion relation $E_n^+$ in place of the quadratic approximation in the denominator of the right hand side of (A.8). This causes the sum over $n$ to diverge logarithmically with the upper limit, leading to a divergent term $\sim \log(E_{\text{max}}/\gamma)/(\hbar\omega_1)$ where $E_{\text{max}} \equiv n_{\text{max}}\hbar\omega_1$. The effect of this term is weak provided $V_0 m_1/(2\pi\hbar^2) \lesssim 1/\log(E_{\text{max}}/\gamma)$, or equivalently, using (18), provided $E_B^0/\gamma \lesssim \sqrt{m_2/m_1}(1 + m_2/m_1)/[\log(E_{\text{max}}/\gamma)]^2$. Thus, for typical situations, in which the logarithm will be of order one, it is sufficient to study (A.8) to understand the properties of shallow bound states with energies $E_B^0 \ll \gamma$.

## B  Non-zero temperatures

We provide here further details of the calculations leading to the results presented in Fig. 3. We consider a set of spinless electrons in a system of total area $A$. These can occupy a set of $N_{\mathrm{imp}} = n_{\mathrm{imp}} \times A$ boundstates on the impurities, each at the energy $E_{\mathrm{imp}} = E_*^+ - E_{\mathrm{B}}$, and a set of unbound Landau level states, with energies $E_n$ and each with degeneracy $N_\phi = n_\phi \times A$ where $n_\phi = eB/h$. Thus the grand potential per unit area is

$$\Omega = -k_B T n_{\mathrm{imp}} \ln\left[1 + \mathrm{e}^{-\beta(E_{\mathrm{imp}}-\mu)}\right] - k_B T n_\phi \sum_{n=0}^{\infty} \ln\left[1 + \mathrm{e}^{-\beta(E_n-\mu)}\right], \tag{B.1}$$

where $\mu$ is the chemical potential and $\beta = 1/(k_B T)$ at temperature $T$. We fix the chemical potential $\mu$ by the condition that there is one electron per impurity site $-\partial\Omega/\partial\mu = n_{\mathrm{imp}}$, to determine $\mu$ as a function of temperature.

We compute the magnetization per unit area, $M = -(\partial\Omega/\partial B)_{\mu,T}$, using the expressions for the oscillatory part of the impurity energy (23), and taking the energies of the unbound states $E_n^+$ within the quadratic approximation (13), suitable for states close to the band edge. Focusing on the dominant contribution to the component that oscillates at the fundamental period, one obtains

$$M \simeq \left\{ -n_{\mathrm{imp}} \frac{1}{1 + \mathrm{e}^{\beta(E_{\mathrm{imp}}-\mu)}} 8\pi E_{\mathrm{B}}^0 \sin(2\pi n_*^+) \mathrm{e}^{-2\pi\sqrt{2E_{\mathrm{B}}^0/\epsilon}} \right.$$
$$\left. + n_\phi \sum_{n=-\infty}^{\infty} \frac{\epsilon(n - n_*^+)}{1 + \mathrm{e}^{\beta[E_*^+ + (1/2)\epsilon(n-n_*^+)^2 - \mu]}} \right\} \frac{dn_*^+}{dB}. \tag{B.2}$$

The magnetization oscillates with $n_*^+$ (12), vanishing for $n_*^+$ integer or half-integer. The amplitudes of the contributions from bound impurity levels and from the unbound states are

$$\Delta M_{\mathrm{imp}} = n_{\mathrm{imp}} \frac{1}{1 + \mathrm{e}^{\beta(E_{\mathrm{imp}}-\mu)}} 8\pi E_{\mathrm{B}}^0 \mathrm{e}^{-2\pi\sqrt{2E_{\mathrm{B}}^0/\epsilon}} \frac{(p_*^+)^2}{2eB^2\hbar}, \tag{B.3}$$

$$\Delta M_{\mathrm{unbound}} = n_\phi \sum_{n=-\infty}^{\infty} \frac{\epsilon(n + 1/4)}{1 + \mathrm{e}^{\beta[E_*^+ + (1/2)\epsilon(n-1/4)^2 - \mu]}} \frac{(p_*^+)^2}{2eB^2\hbar}, \tag{B.4}$$

where we have used $dn_*^+/dB = -(p_*^+)^2/(2eB^2\hbar)$.

These quantities are plotted in Fig. 3, both separately and as their sum, for illustrative parameters $\epsilon = E_{\mathrm{B}}^0$ and $n_\phi = 2n_{\mathrm{imp}}$. At fixed $n_{\mathrm{imp}}$ this latter condition only holds exactly at a specific value of the magnetic field, whereas the field must be varied in order to see quantum oscillations. However, the fractional change of field $\Delta B/B = 2eB/(p_*^+)^2 = 1/n_*^+$ that leads to a full period of oscillation of the magnetization, $\Delta n_*^+ = 1$, is typically small in the regime of interest for quantum oscillations, where the Landau level index at the extremum is large, $n_*^+ \gg 1$. Thus, the condition $n_\phi = 2n_{\mathrm{imp}}$ should be interpreted in the sense that it is applied within the narrow range of fields $\Delta B/B \sim 1/n_*^+$ required to observe a few oscillations.

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
