# Peer review of "Quantum Oscillations in an Impurity-Band Anderson Insulator"

_SciPost Physics, doi:SciPost Phys. 15, 118 (2023)_

## Round 1 · Referee Report · Anonymous (Referee 1) · 2023-6-14

Strengths

1. The paper is clear and didactic
2. The topic is important and of current interest
3. The derived results are significant

Report

The realisation over the past decade that quantum oscillations are not only seen in metals has rewritten the received wisdom of a long-established field of research. This paper, co-authored by one of the pioneers of the theoretical approaches that have been adopted, reports on an important extension of the theory to the case of Anderson localised impurity bands. The paper is beautifully written, and will be of considerable interest to both experimentalists and theorists. I warmly recommend publication.

Requested changes

The only change I recommend is correction of a typo on page 10: 'separately' should be 'separated'.

  • validity: high
  • significance: high
  • originality: high
  • clarity: top
  • formatting: excellent
  • grammar: excellent

Author:  Nigel Cooper  on 2023-08-03  [id 3866]

(in reply to Report 1 on 2023-06-14)
Category:
remark

We thank the referee for this positive and supportive report, and for their careful reading that picked up this typo. (Now corrected.)

---

## Round 1 · Referee Report · Anonymous (Referee 2) · 2023-6-28

Report

A full, satisfactory understanding of the experimental data in Kondo insulators and insulating semiconductor quantum wells still remains elusive, however, the fact that insulators exhibit quantum oscillations (QOs) is by now firmly established. Early work of the senior author, Cooper, with Knolle [20, 29], gave a beautifully simple explanation of how this might occur, without invoking any exotic states of matter.

In this paper, the authors generalize their results to include the effects of disorder going beyond the self-energy analysis of refs. [32,33] and taking into account the localized nature of the impurity bound states. This approach is possibly motivated by the work of Skinner [55], who was the first to emphasize the unusual nature of impurity bound states in insulators where the band edge extrema lie on rings (or surfaces) rather than points.

The authors consider a simple model of a very short-range impurity potential, find the resulting bound state energy and show that its magnetic field variation also leads to a 1/B-periodic contribution to the magnetization. This is in addition to the band insulator (BI) oscillations [20] with a different characteristic field damping. The authors then briefly discuss the implications of their results for experiments and conclude that these impurity-induced oscillations are likely to be more important in the semiconductor quantum wells rather than Kondo insulators.

This manuscript is an important contribution to the field, and definitely merits publication in SciPost. That said, I have several questions that I would like the authors to address better in a revised version.

(1) I find the use of the words ``Anderson insulator’’ somewhat overly simplistic given that the authors actually consider a *single* impurity problem. A very dilute set of randomly located impurities, with essentially no overlap of bound state wave functions, leads to an almost trivial Anderson insulator with localization length equal to the bound state radius (which the authors call) a_0.

It was argued in ref. [55] that the usual Mott criterion for a metal-insulator transition is evaded due to the nature of bound states in this problem, and the system remains insulating in 3D even for a reasonably dense set of impurities. But then the resulting Anderson insulator may have a localization length significantly larger than a_0, which could have important implications (see further below).

(2) I was not clear about the length scales involved in the field suppression of the quantum oscillations. Let me raise several questions related to this.

a) Eq. (27) is interpreted in terms of a flux quantum through an area of (a_0 \lambda_{*}^{+}). But the physical meaning of this area is quite unclear since both the lengths describe a feature of the bound state w.f. in the same (radial) direction, one of them the slow decay and the other the fast oscillations in eq. (20), so their product does not describe a physical area.

b) In Eq. (28), the algebra does lead to an expression involving the ratio of cyclotron radius R_c to the bound state size a_0. But is there a physical picture of electronic motion in cyclotron orbits involved in the calculation that the authors present? Or is it just the field dependence of the energy?

c) If cyclotron orbits are involved, then should one be comparing R_c to the localization length rather than a_0, which might make a big difference in the damping estimate of Eq. (28).

d) In the recent STM paper by Pirie et al. in Science (2023) [61] R_c ~ 100 nm while the size of the metallic puddles (crudely the “localization length”) is ~ 3 nm. So wouldn’t your eq. (28) suggest that you would get an extreme suppression, i.e., the metallic puddles seen by STM could not be responsible for the quantum oscillations. If the authors agree, they should say this much more clearly in their paper.

3) The authors find that impurity level QOs have frequencies set by p_{*}^{+} and p_{*}^{-} in the magnetization, similar to the DOS oscillations [33], rather than the p_{*} scale of band-insulator QOs [20].

This suggests the following simple-minded picture. It is the band edge that oscillates periodically with (1/B), as is known from the DOS calculations, and the binding energy relative to this oscillating band edge is essentially constant, equal to the B=0 value with a possibly smooth B-variation. This would make the energy of the impurity level have an oscillatory component. Is this picture sensible? The authors should consider commenting on this.

4) I was not very clear how the plot in Fig. 3 was obtained. How was the amplitude of the QOs obtained as a function of T? The theory shown is all at T=0, so a few details might be useful. Also, I did not understand the condition n_\phi = 2 n_{imp} because to see the QOs one has to look at the M(B) for a range of B’s, so one cannot have a fixed n_\phi?

In conclusion, the research presented is of a very high quality and the results will be of great interest to many colleagues who would like to understand better the mysteries of quantum oscillations in insulators. After the authors have satisfactorily addressed the questions raised above, I would be very happy to recommend publication in SciPost.

  • validity: top
  • significance: top
  • originality: top
  • clarity: top
  • formatting: excellent
  • grammar: excellent

Author:  Nigel Cooper  on 2023-08-03  [id 3867]

(in reply to Report 2 on 2023-06-28)
Category:
answer to question

The referee writes

A full, satisfactory understanding of the experimental data in Kondo insulators and insulating semiconductor quantum wells still remains elusive, however, the fact that insulators exhibit quantum oscillations (QOs) is by now firmly established. Early work of the senior author, Cooper, with Knolle [20, 29], gave a beautifully simple explanation of how this might occur, without invoking any exotic states of matter.

In this paper, the authors generalize their results to include the effects of disorder going beyond the self-energy analysis of refs. [32,33] and taking into account the localized nature of the impurity bound states. This approach is possibly motivated by the work of Skinner [55], who was the first to emphasize the unusual nature of impurity bound states in insulators where the band edge extrema lie on rings (or surfaces) rather than points.

The authors consider a simple model of a very short-range impurity potential, find the resulting bound state energy and show that its magnetic field variation also leads to a 1/B-periodic contribution to the magnetization. This is in addition to the band insulator (BI) oscillations [20] with a different characteristic field damping. The authors then briefly discuss the implications of their results for experiments and conclude that these impurity-induced oscillations are likely to be more important in the semiconductor quantum wells rather than Kondo insulators.

This manuscript is an important contribution to the field, and definitely merits publication in SciPost. That said, I have several questions that I would like the authors to address better in a revised version.

Our response We thank the referee for their positive and supportive report, and for raising these interesting and useful queries. We provide answers below, and have modified the manuscript to clarify the issues raised. (We attach a pdf file that highlights the changes made.)

The referee writes

(1) I find the use of the words Anderson insulator’’ somewhat overly simplistic given that the authors actually consider a single impurity problem. A very dilute set of randomly located impurities, with essentially no overlap of bound state wave functions, leads to an almost trivial Anderson insulator with localization length equal to the bound state radius (which the authors call) $a_0$.

It was argued in ref. [55] that the usual Mott criterion for a metal-insulator transition is evaded due to the nature of bound states in this problem, and the system remains insulating in 3D even for a reasonably dense set of impurities. But then the resulting Anderson insulator may have a localization length significantly larger than $a_0$, which could have important implications (see further below).

Our response Indeed ,the analysis given is restricted to a regime of dilute impurities. However, this limit is sufficient to establish that quantum oscillations will persist in the Anderson insulator. For clarity we have emphasised this point, and the limitation of the detailed analysis, early in the revised paper. In light of this query (and the remarks of the referee below), we have also added a short discussion of the expected modifications away from the dilute limit.

The referee writes

(2) I was not clear about the length scales involved in the field suppression of the quantum oscillations. Let me raise several questions related to this.

a) Eq. (27) is interpreted in terms of a flux quantum through an area of ($a_0 \lambda_{\star}^{+}$). But the physical meaning of this area is quite unclear since both the lengths describe a feature of the bound state w.f. in the same (radial) direction, one of them the slow decay and the other the fast oscillations in eq. (20), so their product does not describe a physical area.

Our response Indeed, the bound states are rotationally symmetric, so these are both radial length scales: The wavefunction has overall extent $a_0$ with phase alternating in sign on a length scale of $\lambda_\star^+$. The quantity mentioned ($a_0 \lambda_\star^+)$ can be viewed as the “typical" area over which the wavefunction is constant (or of constant sign): a ring of circumference $2\pi a_0$ and width $\lambda_{\star}^{+}$. We have added this remark to the manuscript.

The referee writes

b) In Eq. (28), the algebra does lead to an expression involving the ratio of cyclotron radius $R_c$ to the bound state size $a_0$. But is there a physical picture of electronic motion in cyclotron orbits involved in the calculation that the authors present? Or is it just the field dependence of the energy?

Our response The relevance of the cyclotron orbit arises because the wavefunction of the shallow bound state is dominated by the free-particle states at the energy minimum of the upper band. These are Landau level wavefunctions with (integer) indices close to $n_\star^+$. The amplitude of these free-particle states is concentrated on a ring of radius $R_c = \ell_B \sqrt{2 n_\star^+} \sim p_\star^+/(eB)$.

Thus, physically, it is natural to compare this free-particle state to the size of the bound state. If the bound state is much larger than $R_c$, then the free-particle states are largely unaffected by the binding (oscillations are strong). If the bound state is much smaller than $R_c$, then the free-particle states are suppressed.

Another way to obtain the comparison is via the uncertainty principle in momentum space. In a disorder-free system, the semiclassical orbits in momentum space are separated by an area of $2\pi(\hbar^2/\ell_B^2)$. Here the relevant states (close to the band edge) are circles of radius close to $p_\star^+$, and their separation in radius is $\Delta (p) = \hbar/R_c$. Localisation to a scale $a_0$ leads to an uncertainty of radial momentum of $\sim \hbar/a_0$. Thus, for $\hbar/a_0 \gtrsim \Delta(p/\hbar)$ the semiclassical quantization is washed out, and hence one expects suppression of the quantum oscillations.

We have added these discussions to the revised paper.

The referee writes

c) If cyclotron orbits are involved, then should one be comparing $R_c$ to the localization length rather than $a_0$, which might make a big difference in the damping estimate of Eq. (28).

Our response As commented above, indeed, the detailed calculations are in the regime where the localisation length is $a_0$. For denser impurities, the single particle wavefunctions could extend beyond $a_0$, through tunnelling between the bound states on different impurities. A full analysis of this regime is complex, due to a number of competing length scales and regimes that one can consider, and is beyond the scope of the present paper. However, we add a comment on how the formulas are likely modified. In particular, we believe that the relevant comparison is not with the localisation length, but with the mean-free path of the particle, which sets the uncertainty in momentum. This accounts for the suppression of quantum oscillations even in regimes of metallic behaviour, where the localisation length diverges, and connects with the usual Dingle formula in this regime.

The referee writes

d) In the recent STM paper by Pirie et al. in Science (2023) [61] $R_c$ ~ 100 nm while the size of the metallic puddles (crudely the “localization length”) is ~ 3 nm. So wouldn’t your eq. (28) suggest that you would get an extreme suppression, i.e., the metallic puddles seen by STM could not be responsible for the quantum oscillations. If the authors agree, they should say this much more clearly in their paper.

Our response That is the correct. We have added a comment to emphasize this in the revised paper.

The referee writes

3) The authors find that impurity level QOs have frequencies set by $p_{\star}^{+}$ and $p_{\star}^{-}$ in the magnetization, similar to the DOS oscillations [33], rather than the $p_{\star}$ scale of band-insulator QOs [20].

This suggests the following simple-minded picture. It is the band edge that oscillates periodically with $(1/B)$, as is known from the DOS calculations, and the binding energy relative to this oscillating band edge is essentially constant, equal to the $B=0$ value with a possibly smooth $B$-variation. This would make the energy of the impurity level have an oscillatory component. Is this picture sensible? The authors should consider commenting on this.

Our response This is the correct way to view the oscillations of the impurity level in the limit of very weak attractive potential, for which the binding energy goes to zero. Then, indeed, the oscillations are (trivially) those of the band edge. For nonzero binding energy, these oscillations are suppressed below those of the band edge in the (exponential) manner described in the paper. This suppression is not just a smooth $B$-variation, since it applies at the oscillation frequency. We have added a comment on this to the paper, as we agree that it is useful to make connection to the oscillations of the band edge as the referee suggests.

The referee writes

4) I was not very clear how the plot in Fig. 3 was obtained. How was the amplitude of the QOs obtained as a function of $T$? The theory shown is all at $T=0$, so a few details might be useful. Also, I did not understand the condition $n_\phi = 2 n_{imp}$ because to see the QOs one has to look at the $M(B)$ for a range of $B$’s, so one cannot have a fixed $n_\phi$?

Our response We have added a new Appendix that includes the formulas used, and addresses all of these points. To see quantum oscillations, one only has to look at $M(B)$ for a relatively small range of $B$’s, so it still makes sense to consider the condition $n_\phi = 2 n_{imp}$. In preparing this new Appendix we noticed a small error in the numerical calculation which is corrected in the updated paper through a revised Fig 3.

The referee writes

In conclusion, the research presented is of a very high quality and the results will be of great interest to many colleagues who would like to understand better the mysteries of quantum oscillations in insulators. After the authors have satisfactorily addressed the questions raised above, I would be very happy to recommend publication in SciPost.

Our response We thank the referee again for their careful reading of the paper, and for raising these helpful questions. We believe that the additions have helped to improve the clarity of the paper, and hope that the referee feels that we have addressed their questions adequately.

Attachment:

Cooper_Kelsall_Scipost_diff.pdf

---

## Round 2 · Referee Report · Anonymous · 2023-8-3

Report

The authors have answered all my questions satisfactorily in their response and revised their paper accordingly. The revise manuscript should be accepted for publication.

---

## Round 2 · List of Changes

Changes are highlighted in the pdf file attached to response to Report 2.

---

## Editorial Decision

published